# Numerical Investigation of the Impact of Cracks and Stiffness Loss in the Supporting System for the Dynamic Characteristics of a Rotating Machine

**DOI:** 10.3390/ma17225444

**Published:** 2024-11-07

**Authors:** Grzegorz Żywica, Jan Kiciński

**Affiliations:** Institute of Fluid-Flow Machinery, Polish Academy of Sciences, 80-231 Gdańsk, Poland

**Keywords:** rotor dynamics, rotor vibrations, supporting structure, dynamic stiffness, bolt damage, foundation stiffness

## Abstract

In the literature on rotating machinery, many articles discuss the analysis of various rotor and bearing defects, including both sliding and rolling bearings. Defects in the rotor supporting system are investigated much less frequently. In rotor-bearing-supporting structure systems, where there are couplings between the individual sub-systems, damage to the supporting structure can significantly impact the dynamic properties of the entire machine. The authors of this article have, therefore, focused on analysing the defects that can occur in the supporting system of the rotor and bearings. This article presents the results of a numerical analysis of two common defects in the supporting structure: cracks in the bolted joints attaching the machine body to the foundation and a decrease in foundation stiffness. The research object was a test rig that accurately reproduced the dynamic phenomena occurring in rotating machinery, such as vapour and gas turbines. In the numerical model of the rotating machine, a three-dimensional linear model of the supporting structure was combined with a beam model of the rotor line via a nonlinear fluid film-bearing model. The developed model allowed for the analysis of two different failures in the supporting system over a wide range of rotational speeds. The calculations showed that damage to the supporting structure can significantly impact the dynamic characteristics of the entire rotating machine.

## 1. Introduction

Defects in rotating machinery are often the subject of research by those involved in its design, operation, and diagnostics. A large number of studies on these issues can be found in the literature from the past few decades. Most studies, however, focus on defects in components directly related to rotary motion, such as rotors, couplings, gears, or bearings [1]. Much less attention is paid to the components of the rotor supporting structure, such as casings, vibration isolators, or foundations, which are also essential for the proper operation of the machine.

Studies on vapour turbines, which are a typical example of rotating machinery, indicate that defects in the rotor supporting system are a significant and common problem [2,3]. According to some studies, damage to supporting structure components is responsible for approximately 10% of failures in turbine sets [3]. These typically include defects such as various external and internal cracks in the casings, the loosening of attachments for the casings and bearing supports, cracks in the supports and foundations, and the settlement of the foundation [2]. Therefore, these defects often pose a serious problem. This is because the rotating machine is a single coherent system with couplings between its individual sub-systems [4,5,6]. Severe damage to one component of the rotor-bearing-supporting structure can, therefore, lead to a deterioration in the operating conditions of the entire system. In this context, defects in the supporting structure also need to be analysed thoroughly and extensively, which can contribute to a better assessment of the technical condition of rotating machinery and help avoid many failures.

Taking the supporting structure into account in the analysis of the rotating system allows for an accurate representation of the dynamic phenomena occurring in rotating machines, which is particularly important in the case of elastic supports [7,8]. The influence of the supporting structure on the vibrations of rotating elements is of great importance not only in the case of flexible, multi-support rotors, commonly used in steam and gas turbines and as propulsion shafts [9], but also in machines such as wind turbines [10] and tidal turbines [11]. The vibrations of the supporting system should also be considered in the case of rotating machines exposed to vibrations related to the working environment. Examples of such machines include offshore wind turbines [12] and aircraft engines [13]. In some cases, it is also essential to consider overloads acting on the supporting system or manufacturing errors that may cause bearing misalignment [14] or excessive clearance [9]. Such imperfections should also be taken into account when analysing the vibrations of rotors.

The dynamic characteristics of the supporting system can be obtained experimentally, analytically [15], or through an appropriately prepared numerical model [16] and can be included in analyses conducted in the time or frequency domain [17]. Various methods of analysing rotating machines with the support system taken into account are proposed in the literature [15]. It can be achieved, for example, by transforming the dynamic characteristics of the bearing supports into actual matrices of complex stiffness, damping, and mass coefficients or by using the mixed coordinates method [18]. The choice of an appropriate method for modelling the interactions between components and the degree of reduction in complex dynamic systems depends on many factors, including the required accuracy and calculation speed. The reduction in the number of degrees of freedom and the linearisation of nonlinear systems allows for a reduction in computation time but simultaneously reduces the quality and usefulness of the results obtained [19]. Therefore, in many cases, it is reasonable to combine linear and nonlinear models [20,21], which allows for the most relevant physical phenomena to be taken into account with an acceptable computation time. An example of such a fluid-flow machine model was used in the research discussed later in this article.

The remainder of this article discusses a numerical study aimed at assessing the effect of selected defects in the supporting structure on the dynamic properties of the rotating machine. Two types of failure are considered: a crack in the bolted joints that secure the body to the foundation and a decrease in the stiffness of the foundation. This combination of two types of defects, the size and location of which differ significantly, allows for studies to be carried out that enable more general conclusions to be drawn than is the case when comparing the effects of similar defects with one another. The proposed model of the supporting structure and the test method are versatile enough to be applied to the analysis of different classes of defects in the rotor’s supporting system, which distinguishes them from models dedicated to the analysis of only specific types of defects. The research was conducted using a previously verified numerical model of a rotating machine. The proposed fluid-flow machine model is characterised by the combination of a linear model of the supporting structure, which fully represents its geometry and a nonlinear model of the bearings and rotor. The rotating system has been integrated into the supporting structure using a modified dynamic characteristic transformation method proposed by the authors, which is sensitive to even small changes in the dynamic characteristics that directly affect the vibration level of the rotor. To perform a dynamic analysis of the system being studied with the proposed damage, a series of calculations were conducted using both commonly available and in-house-developed software. The proposed approach allowed conclusions to be drawn about the impact of the defects under consideration on the dynamic properties of the rotating machine being studied. As the obtained results show, it is not always the case that extensive damage to the supporting structure negatively affects rotor vibration; conversely, a small defect at a sensitive location can lead to increased vibration levels and unstable machine operation.

## 2. Rotating Machine Model

To conduct numerical tests, a model of the entire rotating machine was developed, taking into account the rotor, bearings, and supporting structure. The research object was a large-scale test rig for rotor and bearing dynamics located in Gdańsk, Poland, at the Institute of Fluid-Flow Machinery of the Polish Academy of Sciences (IMP PAN). A diagram of this test rig is provided in Figure 1.

General-purpose finite element analysis software Abaqus/CAE v2016 [22] was used to analyse the rotor supporting structure, which has a mass of approximately 13,000 kg. A detailed discussion of the modelling process and model verification was the subject of an earlier publication [23]. At this point, however, it is important to mention the key features of the developed model (Figure 2). It is a linear model consisting of nearly 400,000 degrees of freedom. Finite elements with first- and second-order shape functions were used to discretise it. At the connection points of the individual components of the supporting structure (e.g., between the concrete block and the steel profiles and at the bolted joints), a so-called rigid contact was used. This means permanently connecting the nodes of the FEM mesh at the points where the different components adjoin. Such a simplification allows for numerical procedures dedicated to linear systems to be used and for calculations to be carried out in the frequency domain. The experimental verification of the model showed a high degree of agreement between the simulated and experimental dynamic characteristics of the bearing supports, as discussed in earlier publications [23,24].

To investigate the influence of selected defects in the supporting structure on the dynamic state of the rotor, appropriate modifications were purposefully introduced into the model shown in Figure 2. These issues will be discussed in more detail in the following section of the article.

The analysis of the rotor, including the fluid film bearings, was carried out using selected programs from the MESWIR series [2]. For the kinetostatic calculations, the KINWIR-60 program, based on a diathermic sliding bearing model, was used. The NLDW-80 program was used to perform the nonlinear dynamic calculations. The two applications complemented each other, as some of the values determined in KINWIR-60 were then used as input parameters for the NLDW-80 program (e.g., support reactions, oil temperature, etc.). To analyse the entire rotor-bearing-supporting structure system over a wide range of rotational speeds while accounting for various failures, it is necessary to use a nonlinear description of the dynamic phenomena that occur in the bearings.

The discrete rotor model developed for the NLDW-80 program is shown in Figure 3. It is a model of a rotor with a diameter of 0.1 m and a length of 3.4 m, comprising two shafts connected by a rigid coupling. Two discs, each with a mass of 185 kg, were mounted on the rotor. The discrete model consists of 29 beam elements (Timoshenko model), with four degrees of freedom at each node. Unbalances were modelled at the locations where the discs were mounted, corresponding to a 15 µm deviation in their principal axes of inertia. These unbalances were rotated 180° relative to one another. The rotor was supported by three fluid film bearings, each with a cylindrical clearance and two lubrication pockets. The radial clearance of the bearings was approximately 80 µm, and they were lubricated with machine oil supplied at a pressure of 0.22 MPa.

Programs from the MESWIR series used for rotor and fluid film bearing calculations have been employed at IMP PAN in Gdańsk and have been experimentally verified numerous times, both in laboratory conditions and on large energy machines [2]. This verification confirmed their high utility for analysing different types of rotating machinery.

The integration of the three-dimensional model of the supporting structure with the beam model of the rotor line and fluid film bearings was achieved through stiffness and damping coefficients in the KINWIR-60 program (C and D in Figure 3) and additionally, through virtual mass coefficients in the NLDW-80 program (M in Figure 3). These coefficients characterised the dynamic properties of the entire supporting structure at the rotor’s support points. In general, it can be stated that they are determined using the dynamic compliance of the supporting structure, obtained by applying a harmonic force at the bearing supports in the horizontal and vertical directions perpendicular to the shaft axis. The location of the force application at the second support is shown in Figure 2. The amplitude of the harmonic force does not affect the value of the dynamic compliance determined because its calculation takes into account the displacement, which, for linear systems, is proportional to the load. The dynamic compliance of the system is determined separately for each frequency of interest. In the case discussed, the dynamic compliance was determined over the frequency range from 1 Hz to 180 Hz, with a resolution of 1 Hz. The dynamic compliance matrices obtained in this manner were subsequently inverted. This mathematical operation yields complex dynamic stiffness matrices. Then, using the two-point method, the desired values of the stiffness (C), damping (D), and virtual mass (M) coefficients were obtained. A detailed discussion of the proposed procedure for determining these coefficient values is included in earlier publications [24,25]. This method was used to transform the dynamic characteristics of the bearing supports into the actual matrices of complex stiffness, damping, and mass coefficients, allowing for even subtle changes caused by defects in the rotor’s supporting structure to be taken into account. This enabled a consistent analysis of the system under study and an assessment of the impact of the considered defects on the dynamic properties of the rotating machine to be analysed.

## 3. Analysis of Defects in the Supporting Structure

To carry out simulation calculations, including defects in the supporting structure, it was necessary to modify the base model. The model of the supporting structure without defects, as previously mentioned, had been verified, and the obtained characteristics were consistent with the experimental results. This model was then used to analyse two different types of defects. These included cracks in the bolts securing the machine body to the foundation block and a decrease in the stiffness of the foundation on which the entire structure was placed. These defects are quite common in practice, so their analysis is justified not only for theoretical reasons but also for practical ones.

### 3.1. Modelling of Cracks in Bolted Joints

Before introducing defects in the bolted joints into the model of the supporting structure, an analysis of the effect of bolt tension on the dynamic properties of the system was conducted. The Abaqus/CAE system used includes built-in tools for performing such calculations [22,26,27]. The results obtained confirmed only a minor influence of this factor on both the natural frequencies and mode shapes of the supporting structure, as well as on the dynamic characteristics of the bearing supports. The maximum differences in natural frequencies were as low as 0.01 Hz. In further calculations, the tension in the bolted joints was not considered. However, it should be noted that the base model uses a ‘rigid’ contact between the elements of the bolted joint and the connected parts, which ensures the fixed relative positions of these elements.

To carry out the dynamic calculations of the rotor, it is first necessary to obtain the dynamic compliance waveforms for the bearing supports, from which the stiffness, damping, and mass matrices are determined and then introduced into the rotor programs. For this purpose, the first stage of the study involved calculations of the supporting structure itself in the presence of the introduced defects. Figure 4 shows the location of the defects in the form of cracks in the bolts connecting the two parts of the machine. During the simulation studies, two variants of the cracks were examined. The first variant involved a crack in one bolt, while the second variant involved cracks in two bolts connecting the machine’s body to the foundation. The bolt crack, in the form of a permanently open gap, was modelled using the ‘Crack-Assign Seam’ option in the Abaqus/CAE system. This software places overlapping duplicate nodes along a seam when the mesh is generated, and the applied crack model has a top corner radius of zero. Such a permanently open gap that ignores contact phenomena does not disrupt the system’s linearity, thus allowing the modal superposition method to be used to determine the dynamic compliance of the bearing supports. The results of the simulation calculations, presented as complex dynamic compliance waveforms in the horizontal (L_33_) and vertical (L_44_) directions, are shown in Figure 5 and Figure 6.

These figures clearly show that the introduced defects significantly affected the characteristics of the second bearing support. The largest changes in the characteristics compared to the model without defects were observed in the compliance in the vertical direction. At excitation frequencies close to 30 Hz (specifically at 35 Hz in the first case and at 29 Hz in the second), a distinct anti-resonance emerged. At higher frequencies, however, the curve exhibits a ‘smoothing’ effect, and the local anti-resonances present in the base model at frequencies of 68 Hz and 99 Hz diminish. In the horizontal direction, changes in dynamic compliance were observed at frequencies above 8 Hz. In the first bolt crack variant, the frequency of the highest anti-resonance decreased from 31 Hz to 29 Hz, while in the second variant, it decreased to 24 Hz. In both crack variants, the introduction of defects led to an increase in dynamic compliance in both the horizontal and vertical directions across nearly the entire frequency range.

### 3.2. Modelling of a Drop in Foundation Stiffness

The second defect considered in the rotor supporting structure was a decrease in foundation stiffness. To carry out the appropriate calculations, the following modification was made to the supporting structure model: the unilateral restraint of the elements modelling pneumatic vibration isolators was replaced with elastic elements. This modification involved elements beneath half of the structure, which could correspond to, for example, an uneven foundation settlement. For this defect, two variants were also considered: stiffness in three directions was set to 1 MN/m and 0.5 MN/m. Figure 7 shows the numerical model with defects introduced as a decrease in foundation stiffness.

As can be seen from the Figure 8 and Figure 9, the largest changes compared to the base characteristics were recorded at low excitation frequencies. In this range, both the frequency of resonances and the corresponding values of the modulus of complex dynamic compliance changed. The maximum values of dynamic compliance in the horizontal direction, which originally occurred at 4 Hz and 8 Hz, shifted by 1 Hz towards lower frequencies. Similar changes occurred in the dynamic compliance curve in the vertical direction, where the maximum value of this parameter, originally at 8 Hz, shifted to 6 Hz in the first case and 5 Hz in the second after the defect was introduced. This is because the decrease in ground stiffness led to changes in the modal model of the system, affecting low-frequency mode shapes that correspond to the oscillations of the entire structure as a rigid body. The remaining characteristics, corresponding to higher excitation frequencies, remained virtually unchanged.

The next section of the article will present the calculation results obtained for a rotor mounted on a supporting structure with the defects discussed in the following section. The complex dynamic compliance waveforms of the bearing supports, obtained as discussed above, will be used to determine the parameters required for analysing the rotor. Indeed, rotor programs require the complex stiffness, damping, and mass coefficients that characterise the properties of the supporting structure at the bearing support locations.

## 4. Effect of Defects in the Supporting Structure on Rotor Vibrations

Some of the defects analysed in the previous section of this article significantly altered the dynamic properties of the rotor’s supporting structure. In this section, analyses are presented to assess how these changes affected the dynamic characteristics of the rotating system and, consequently, the vibrations of the entire rotating machine.

Figure 10 shows the numerically determined waveforms of the journal-sleeve relative vibration amplitude for a rotating machine model without defects (base case). This is a characteristic typical of a rotor supported by fluid film bearings. A distinction can be made here between a region of stable rotor operation, with a distinct resonance occurring at a speed of approximately 2900 rpm, and the limitations of stability in the system occurring at a speed of approximately 4200 rpm. Beyond this limit, oil whirls appear, rapidly transitioning to oil whip, which causes a sharp increase in the vibration level. The characteristic shown in Figure 10 serves as a reference for the waveforms of the journal-sleeve vibration amplitude obtained for rotating machine models with introduced defects.

### 4.1. Rotor Vibration with Cracks in the Supporting Structure

Figure 11 and Figure 13 show the relative vibration amplitude waveforms obtained for a rotating machine model with defects introduced as cracks in the bolts securing the machine body to the foundation block. In the first case, where a crack in one bolt was simulated (Figure 11), the most significant change compared to the base model characteristics was a noticeable shift in the system’s limitations of stability to higher rotational speeds. The oil whirls, in this case, appeared only at approximately 4500 rpm, followed by a slow development until they transitioned to a distinct oil whip, which, in this case, occurred only at around 4900 rpm. To accurately trace the development of oil whirls, a detailed analysis of the vibration trajectories of the bearing journals was required. Figure 12 illustrates the three most characteristic cases of the system’s operation: stable rotor operation (Figure 12a), oil whirls (Figure 12b), and oil whip (Figure 12c). The shift in the stability limit of the system towards higher rotational speeds was, in this case, caused by significant changes in the complex dynamic compliance of the central bearing support in the vertical direction (Figure 5). As a result of these changes, different values for the stiffness, damping, and virtual mass coefficients (which characterise the dynamic properties of the supporting structure) were substituted into the global equation of motion for the system.

Figure 13 shows the relative vibration amplitude waveforms obtained for the model with cracks in the two bolts connecting the machine body to the foundation block (Figure 4). The characteristics obtained in this case resemble those obtained with a crack in one bolt (Figure 11). It should be noted, however, that an additional distinct resonance appeared in the second (middle) bearing at a rotational speed of approximately 4000 rpm. The reason for this is likely to be a significant increase in the dynamic compliance of the second bearing support at higher excitation frequencies (Figure 6) in both the horizontal and vertical directions. In addition, an irregular course of relative vibration amplitude was observed in the second bearing, as well as in the other bearings, at higher rotational speeds. This is due to the previously mentioned changes in the dynamic compliance of the bearing supports.

### 4.2. Rotor Vibration with Reduced Foundation Stiffness

The characteristics presented in Figure 14 and Figure 15 were obtained after introducing modifications to the supporting structure model that corresponded to a decrease in foundation stiffness beneath half of the structure. Detailed analysis revealed that there were no significant changes in the relative vibration amplitude waveforms of the bearing journals within the range of rotational speeds of interest. In both cases considered, the values obtained were practically consistent with those for the base case. This result is as expected, as the defect in the form of a decrease in foundation stiffness—on which the entire structure is seated—caused changes in the dynamic compliance waveforms of the bearing supports only in the low-frequency range, i.e., from several to around a dozen hertz (Figure 8 and Figure 9). This defect is, therefore, unlikely to affect the dynamic properties of the rotating machine under test during normal operation at speeds of several thousand revolutions per minute. It could, however, have a significant impact on the system’s characteristics during the machine’s run-up and run-down.

All the previously discussed changes in the dynamic characteristics of the rotating machine under study, caused by the appearance of defects in the supporting structure, are quite general in nature. This is because each case discussed required a thorough analysis, not only of the final result in the form of the relative vibration amplitude waveforms of the bearing journals but also of the shape of the vibration trajectories, the changes in the complex dynamic compliance and stiffness waveforms of all bearing supports, and the values of the stiffness, damping, and virtual mass coefficients determined from these. As such, detailed analyses are beyond the scope of this article; these issues are not discussed in detail here.

## 5. Conclusions

This article discusses issues related to the modelling of defects in the supporting structure, which is an integral part of any rotating machine. The research method used and the tools employed—comprising both universal and in-house-developed calculation programs—enabled calculations to be performed across a wide range of rotor rotational speeds. The method for transforming the dynamic characteristics of bearing supports, developed specifically for this study, allowed for the consideration of even minor changes in the dynamic properties of the rotor’s supporting structure caused by the appearance of various types of defects.

The results obtained confirmed the significant impact of the proposed defects in the supporting structure on the rotor’s motion parameters and, consequently, on the dynamic characteristics of the entire rotating machine. The largest changes in the relative journal-sleeve vibration amplitude waveforms were recorded for the case of cracks in the two bolts connecting the machine body to the foundation block. Slightly smaller differences, compared to the model without defects, were obtained in the case of a crack in one bolt. In comparison to the base model, the changes mainly involved a shift in the system’s limitations of stability towards higher rotational speeds and the appearance of additional resonances, primarily observed in the middle bearing. The results obtained at higher rotational speeds, after introducing defects in the form of cracks in the bolts, were quite surprising, as this meant that the introduced defects improved the system’s operating conditions. This observation is correct only for certain rotational speeds, where more favourable values were obtained for the stiffness, damping, and mass coefficients describing the dynamic properties of the supporting structure. An important conclusion drawn from the study was the fact that defects in the form of a decrease in foundation stiffness had no significant effect on the dynamic properties of the system within the range of the rotational speeds analysed. This was because a defect of this type only affected the properties of the supporting structure at very low frequencies, i.e., from several to around a dozen hertz. In all cases considered, there were also no significant changes in the occurrence of resonant speed or in the amplitude of journal vibrations at this speed.

The presented research results can be used in the design, operation, and diagnostic stages of rotating machinery. The proposed method for modelling and analysing defects in the rotor’s supporting system can also be used to train artificial neural networks or as a data source for other methods based on artificial intelligence, which are increasingly employed in machine diagnostics and monitoring. Indeed, a similar analysis can be conducted for virtually any rotating machine. The results of the numerical studies obtained in this manner can help prevent many serious failures and prove highly useful when making decisions about the continued operation of the machine. Of course, it is also possible to analyse various other defects in the supporting structure, which will be the subject of the authors’ forthcoming publications.

## Figures and Tables

**Figure 1 materials-17-05444-f001:**
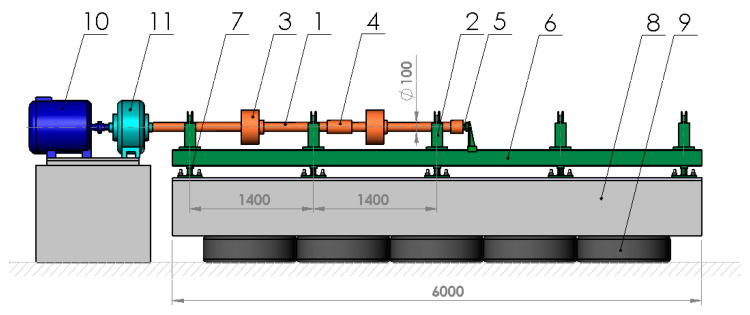
Diagram of the test rig for rotor and bearing dynamics (1—rotor; 2—bearing support; 3—disk; 4—coupling; 5—thrust bearing; 6—steel frame; 7—frame bracket; 8—foundation block; 9—pneumatic vibration isolators; 10—motor; 11—gear transmission).

**Figure 2 materials-17-05444-f002:**
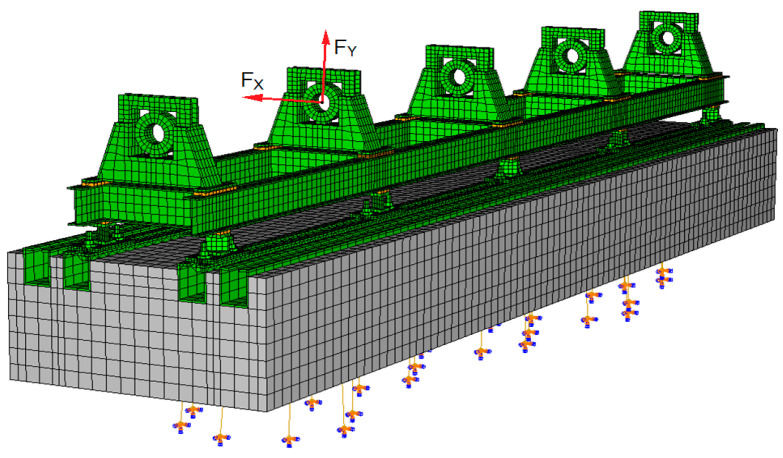
FEM model of the rotor’s supporting structure with forces acting on the second bearing support in the horizontal (F_X_) and vertical (F_Y_) directions.

**Figure 3 materials-17-05444-f003:**
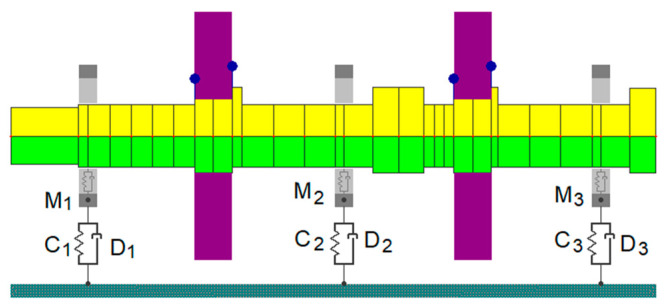
Numerical model of the laboratory rotor (the shaft is marked in yellow and green, and the disks are marked in magenta).

**Figure 4 materials-17-05444-f004:**
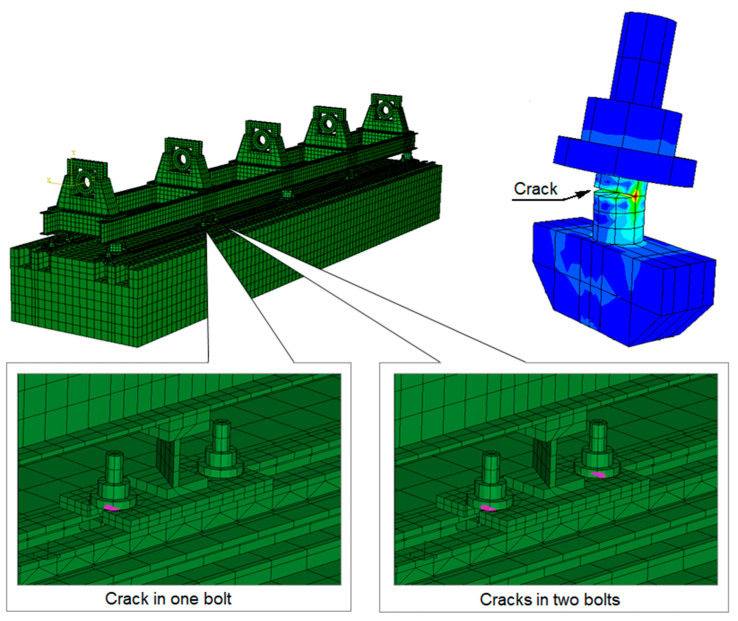
Supporting structure model with defects introduced as bolt cracks.

**Figure 5 materials-17-05444-f005:**
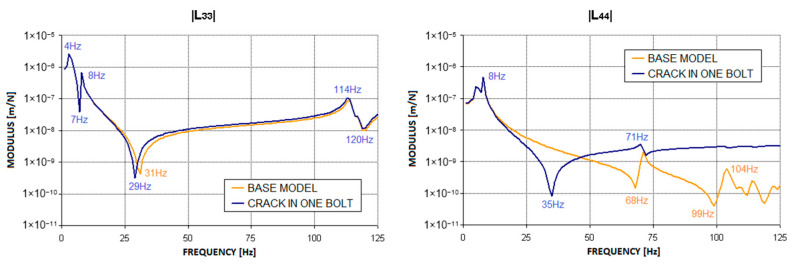
Change in the modulus of the complex dynamic compliance of the second bearing support due to the introduction of a crack in one bolt.

**Figure 6 materials-17-05444-f006:**
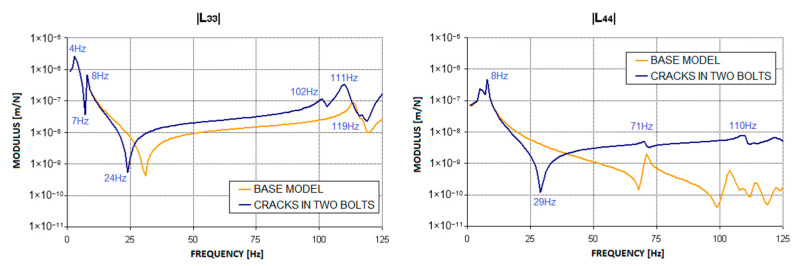
Change in the modulus of the complex dynamic compliance of the second bearing support due to the introduction of cracks in two bolts.

**Figure 7 materials-17-05444-f007:**
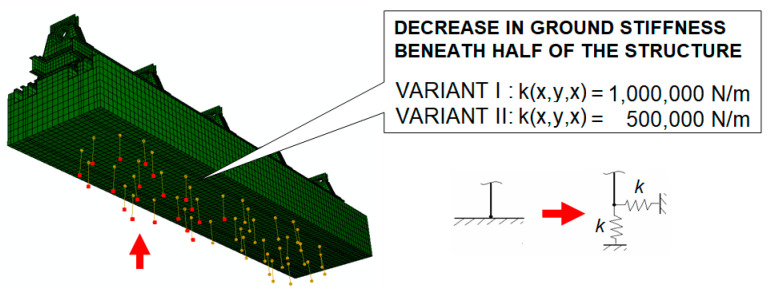
Model of the rotor supporting structure with defects introduced as a decrease in foundation stiffness.

**Figure 8 materials-17-05444-f008:**
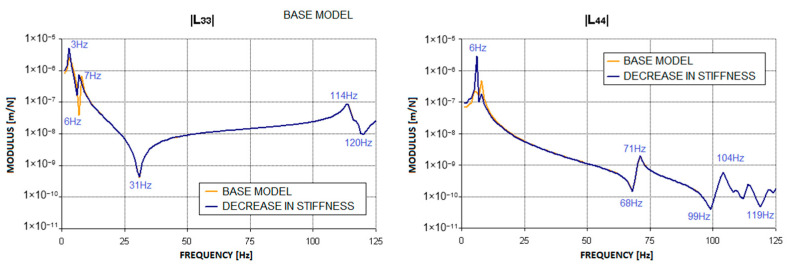
Change in the modulus of the complex dynamic compliance of the second bearing support due to a decrease in foundation stiffness to 1,000,000 N/m.

**Figure 9 materials-17-05444-f009:**
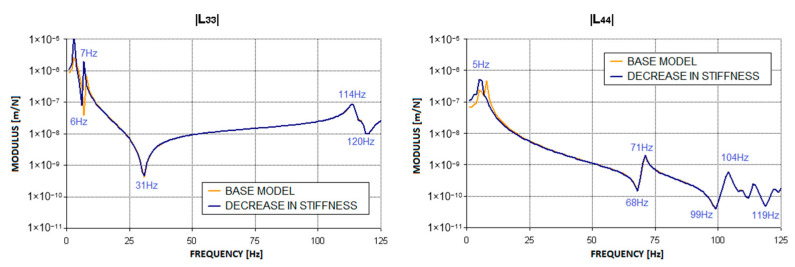
Change in the modulus of complex dynamic compliance of the second bearing support due to a decrease in foundation stiffness to 500,000 N/m.

**Figure 10 materials-17-05444-f010:**
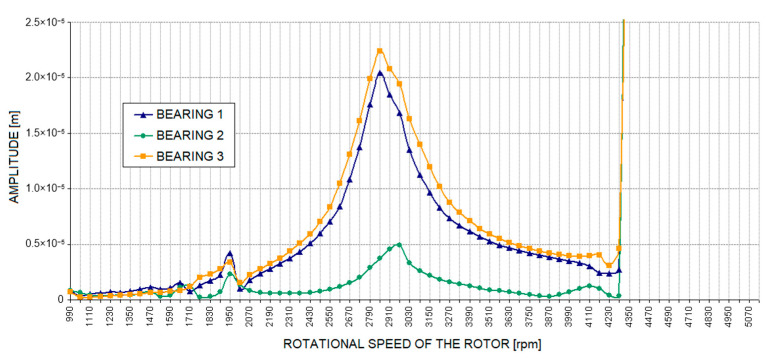
Amplitude of relative journal-sleeve vibrations for the base case.

**Figure 11 materials-17-05444-f011:**
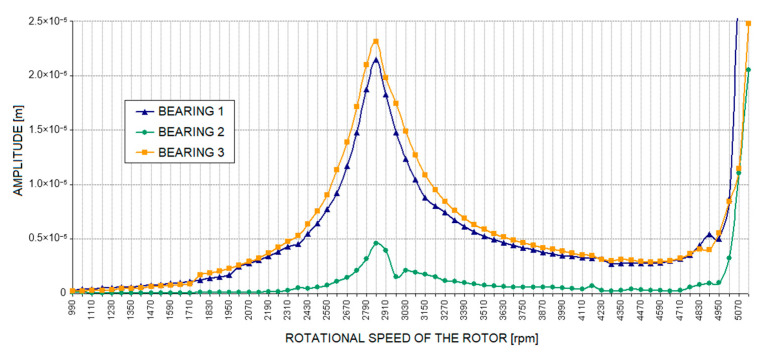
Amplitude of relative journal-sleeve vibrations with a defect in the supporting structure introduced as a crack in one bolt.

**Figure 12 materials-17-05444-f012:**
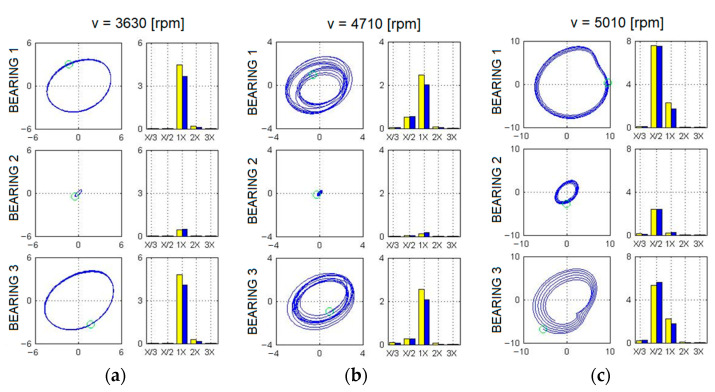
Trajectories of relative journal-sleeve vibrations at different rotational speeds for a model with a defect introduced as a crack in one bolt, showing stable system operations (**a**), oil whirls (**b**), and oil whip (**c**). Yellow bars represent vibrations in the horizontal direction and blue bars represent vibrations in the vertical direction.

**Figure 13 materials-17-05444-f013:**
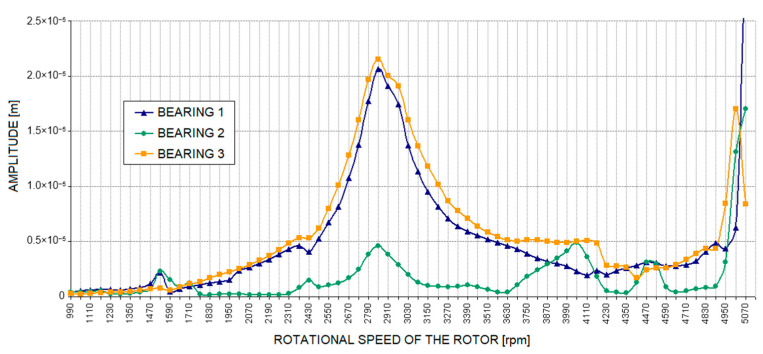
The amplitude of relative journal-sleeve vibrations with a defect in the supporting structure introduced as cracks in two bolts.

**Figure 14 materials-17-05444-f014:**
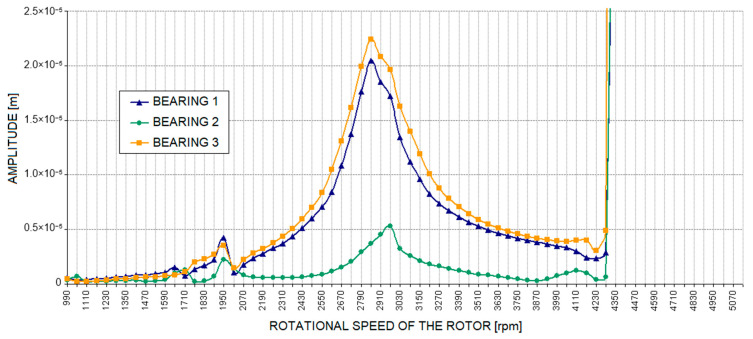
Amplitude of relative journal-sleeve vibrations with a defect in the supporting structure introduced as a drop in foundation stiffness to 1,000,000 N/m.

**Figure 15 materials-17-05444-f015:**
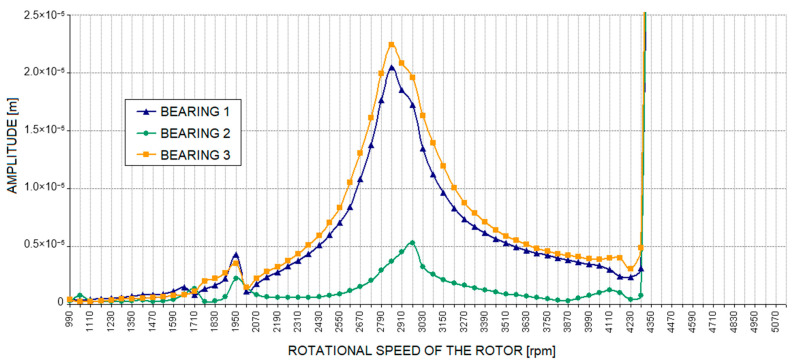
The amplitude of relative journal-sleeve vibrations with a defect in the supporting structure introduced through a drop in foundation stiffness to 500,000 N/m.

## Data Availability

The original contributions presented in the study are included in the article, further inquiries can be directed to the corresponding author.

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
