# Peer review of "Numerical Investigation of the Impact of Cracks and Stiffness Loss in the Supporting System for the Dynamic Characteristics of a Rotating Machine"

_materials, 2024, doi:10.3390/ma17225444_

Round 1
Reviewer 1 Report
Comments and Suggestions for Authors
In this paper, a three-dimensional linear model of the supporting structure is combined with a beam model of the rotor line via a nonlinear fluid film bearing model. The developed model allowed for the analysis of two different failures in the supporting system over a wide range of rotational speeds. The calculations showed that damage to the supporting structure can significantly impact the dynamic characteristics of the entire rotating machine. It is an interesting paper. However, the authors need to address the following issues before this manuscript is accepted:
1. In Introduction, some failure reasons and detection method for the cracks and other defects in rotor system are discussed. It is okay. However, some other failure reasons including the overload and manufacturing errors can also produce the defects, as given in ‘Vibration analysis of the propulsion shaft system considering dynamic misalignment in the outer ring’. Moreover, some vibration methods can also clearly detect the defects, as given in ‘Dynamic modeling and vibration analysis of double row cylindrical roller bearings with irregular-shaped defects’. Those issues can be discussed. The above materials and references can also be discussed.
2. In Figure 3, the M C and D should be explained in the text. Moreover, the modeling method and calculation methods of the C and D should be clearly listed in the text. Because those parameters should greatly affect the vibrations of the rotor system.
3. In section 3.1, the crack is modelled. However, the modelling methods of the crack is missed. Many previous works have studied the effect factors, such as the contacts and crack tips, etc. Those factors should be discussed too.
4. A key issue is that the contact surfaces in the studied system are missed. Is it correct?
5. In results analysis, the key frequencies in the spectra should be discussed and marked in the figures.
Reviewer 2 Report
Comments and Suggestions for Authors
Reviewer's comments:
1. The authors should add a description of the loads used in the model, and could also include them in Figure 2.
2. Sentence in lines 130-133 is confusing and seems unfinished. If the method mentioned here refers to the one described in the previous sentence, then it should start along the lines of "This method was used..."
3. Figure 4 should be slightly increased in size.
4. In figure 12, the authors should add a), b) and c) under the groups of diagrams, to make it clearer which diagrams represent what.
Comments on the Quality of English Language
English is very good throughout the paper, but there are some small corrections that need to be made.
Reviewer 3 Report
Comments and Suggestions for Authors
For the manuscript materials-3201474, this reviewer has the following comments.
The manuscript lacks literature review.
The novelty is not clearly stated.
Effect of defects in the supporting structure has been presented and published by the authors in earlier publications.
To sum up, the manuscript requires significant modification regarding the novelty. It has serious flaws without significant contributions matching the quality standard of this journal.
Therefore, the manuscript is not recommended for publication.
Comments on the Quality of English Languagemoderate proofreadigns
